

# Demographics, perceptions, and socioeconomic factors affecting influenza vaccination among adults in the United States

Kaja M. Abbas[1], Gloria J. Kang[2,3], Daniel Chen[3], Stephen R. Werre[2,4] and Achla Marathe[3,5]

[1] Department of Infectious Disease Epidemiology, London School of Hygiene & Tropical Medicine, London, United Kingdom
[2] Department of Population Health Sciences, Virginia Polytechnic Institute and State University (Virginia Tech), Blacksburg, VA, United States of America
[3] Biocomplexity Institute, Virginia Polytechnic Institute and State University (Virginia Tech), Blacksburg, VA, United States of America
[4] Laboratory for Study Design and Statistical Analysis, Virginia Polytechnic Institute and State University (Virginia Tech), Blacksburg, VA, United States of America
[5] Department of Agricultural and Applied Economics, Virginia Polytechnic Institute and State University (Virginia Tech), Blacksburg, VA, United States of America

Corresponding author
Kaja M. Abbas,
kaja.abbas@lshtm.ac.uk,
kaja.abbas@gmail.com

## ABSTRACT

**Objective**. The study objective is to analyze influenza vaccination status by demographic factors, perceived vaccine efficacy, social influence, herd immunity, vaccine cost, health insurance status, and barriers to influenza vaccination among adults 18 years and older in the United States.

**Background**. Influenza vaccination coverage among adults 18 years and older was 41% during 2010–2011 and has increased and plateaued at 43% during 2016–2017. This is below the target of 70% influenza vaccination coverage among adults, which is an objective of the Healthy People 2020 initiative.

**Methods**. We conducted a survey of a nationally representative sample of adults 18 years and older in the United States on factors affecting influenza vaccination. We conducted bivariate analysis using Rao-Scott chi-square test and multivariate analysis using weighted multinomial logistic regression of this survey data to determine the effect of demographics, perceived vaccine efficacy, social influence, herd immunity, vaccine cost, health insurance, and barriers associated with influenza vaccination uptake among adults in the United States.

**Results**. Influenza vaccination rates are relatively high among adults in older age groups (73.3% among 75+ year old), adults with education levels of bachelor's degree or higher (45.1%), non-Hispanic Whites (41.8%), adults with higher incomes (52.8% among adults with income of over $150,000), partnered adults (43.2%), non-working adults (46.2%), and adults with internet access (39.9%). Influenza vaccine is taken every year by 76% of adults who perceive that the vaccine is very effective, 64.2% of adults who are socially influenced by others, and 41.8% of adults with health insurance, while 72.3% of adults without health insurance never get vaccinated. Facilitators for adults getting vaccinated every year in comparison to only some years include older age, perception of high vaccine effectiveness, higher income and no out-of-pocket payments. Barriers for adults never getting vaccinated in comparison to only some years include lack of health

insurance, disliking of shots, perception of low vaccine effectiveness, low perception of risk for influenza infection, and perception of risky side effects.

**Conclusion**. Influenza vaccination rates among adults in the United States can be improved towards the Healthy People 2020 target of 70% by increasing awareness of the safety, efficacy and need for influenza vaccination, leveraging the practices and principles of commercial and social marketing to improve vaccine trust, confidence and acceptance, and lowering out-of-pocket expenses and covering influenza vaccination costs through health insurance.

## INTRODUCTION

Influenza has caused between 9.2 million and 35.6 million illnesses, between 140,000 and 710,000 hospitalizations, and between 12,000 and 56,000 deaths annually in the United States since 2010 (*CDC, 2017*). Influenza vaccination is an effective public health intervention to lower the morbidity and mortality burden from influenza. The Advisory Committee on Immunization Practices of the Centers for Disease Control and Prevention recommends influenza vaccination for everyone 6 months or older (*Grohskopf et al., 2017*). Influenza vaccination coverage among children (6 months to 17 years) was 59.0% and among adults (18 years and older) was 43.3% during the 2016–2017 influenza season (*CDC, 2018*). While the influenza vaccination coverage among adults 18 years and older has increased from 40.5% during 2010–2011 to 43.3% during 2016–2017, it has plateaued and is below the target of 70% influenza vaccination coverage among adults as part of the Healthy People 2020 initiative (*ODPHP, 2018*).

### Knowledge, attitudes, and beliefs affecting influenza vaccination

Prior studies have identified demographic factors, and knowledge, attitudes and beliefs affecting influenza vaccination, especially among elderly Americans. Demographic factors include age (*Van Essen, Kuyvenhoven & De Melker, 1997*; *Petersen et al., 1999*) and race (*CDC, 2018*). Facilitators include the awareness of the safety, efficacy and need for influenza vaccination (*CDC, 1999*), physician recommendations (*CDC, 1999*; *Zimmerman et al., 2003*), fear of contracting influenza without influenza vaccination (*Zimmerman et al., 2003*), social forces influencing vaccination behavior (*Tabbarah et al., 2005*), and disagreement with the view that the vaccine is detrimental (*Tabbarah et al., 2005*), while barriers include fear of side effects (*CDC, 1999*), efficacy concerns (*Fiebach & Viscoli, 1991*; *CDC, 1999*), and fear of vaccines causing influenza (*Nichol, Lofgren & Gapinski, 1992*; *CDC, 1999*; *Zimmerman et al., 2003*).

### Study objective

The study objective is to analyze influenza vaccination status by demographic factors, perceived vaccine efficacy, social influence, herd immunity, vaccine cost, health insurance status, and barriers to influenza vaccination among adults 18 years and older in the United States.

### Public health significance

The public health significance of this study is to understand and analyze the demographics, perceptions, and socioeconomic factors affecting influenza vaccination among adults in the United States, and provide evidence to improve influenza vaccination rates among the adults toward the Healthy People 2020 target of 70% (*ODPHP, 2018*).

## METHODS

We conducted an online survey in 2016 of a nationally representative sample of the general population of adults 18 years and over in the United States. Data was collected from 2,150 respondents in the survey which was administered by the Gfk Group using a sample from KnowledgePanel, a probabilistic-based web panel designed to be representative of the United States population. The Institutional Review Board at Virginia Tech granted ethical approval for this study (IRB # 14-712), and the survey was conducted by the Gfk Group with the consent of the participants.

### Survey questionnaire

Survey questions were based on the health behavioral framework of the health belief model (*Rosenstock, 1974*; *Coe et al., 2012*; *Santos et al., 2017*) and the socioecological model (*McLeroy et al., 1988*; *Kumar et al., 2012*; *Niyibizi, Schamel & Frew, 2016*). The health belief model illustrates that health-related behavior, such as getting influenza vaccine, is influenced by perceived susceptibility, perceived severity, perceived benefits, perceived barriers, cues to action, and self-efficacy. The socioecological model explains that health-related behavior of individuals is influenced by their perceptions, social influences, and structural factors such as access to health care and insurance. Specifically, our survey questions were focused on understanding perceived vaccine efficacy, social influences and herd immunity, vaccine cost and health insurance, and barriers affecting influenza vaccination in adults, as illustrated in Table 1.

### Bivariate analysis

We analyzed vaccination status by respondent's gender, age, education, ethnicity, income, marital status, metro status, region, work status, and internet access using Rao-Scott's chi-square test (Table 2). We also analyzed vaccination status by perceived vaccine efficacy, social influence and herd immunity, vaccine cost and health insurance, and barriers to vaccination using Rao-Scott's chi-square test (Table 3).

Marital status refers to partnered adults (married or living together) and single adults (never married, separated, divorced, or widowed). Metro status refers to metropolitan areas with a relatively high population density at its centre with proximal economic ties

**Table 1 Influenza vaccination survey.** Factors of vaccination status, perceived efficacy, social influence, herd immunity, vaccine cost, health insurance, and barriers to vaccination were included in the survey questionnaire.

| Factors | Survey questions | Response choices |
|---|---|---|
| Vaccination status | Do you get an influenza vaccine? | 1. Yes, every year<br>2. Yes, some years<br>3. No, never |
| Perceived efficacy | How effective do you think the influenza vaccine is in protecting people from becoming sick with influenza? | 1. Very effective<br>2. Somewhat effective<br>3. Not effective<br>4. It varies from season to season<br>5. Don't know |
| Social influence | Are you more likely to get a vaccine if others around you get a vaccine? | 1. Yes, more likely<br>2. No, less likely<br>3. No, no effect |
| | Are you more likely to get a vaccine if others around you do not get a vaccine? | 1. Yes, more likely<br>2. No, less likely<br>3. No, no effect |
| Herd immunity | Do you get a vaccine to protect yourself, protect others, or protect yourself and others? | 1. Protect myself<br>2. Protect others<br>3. Protect myself and others |
| Vaccine cost | How much do you pay to get an influenza vaccine? | 1. $0<br>2. Less than $30<br>3. $30 to $60<br>4. More than 60$<br>5. Don't know |
| Health insurance | Do you have health insurance? | 1. Yes<br>2. No |
| | Are influenza vaccines covered by your health insurance? | 1. Yes, the full cost is paid<br>2. Yes, but only part of the cost is paid<br>3. No<br>4. Don't know |
| Barriers | What are the reasons you would not get an influenza vaccine? (check all that apply) | 1. The vaccine costs too much<br>2. The vaccine is not very effective in preventing influenza<br>3. I am not likely to get influenza<br>4. Do not know where to get vaccine<br>5. The side effects of the vaccine are too risky<br>6. I am allergic to some of the ingredients in the vaccine<br>7. I do not like shots<br>8. I just don't get around to doing it<br>9. I have to travel too far to get vaccine<br>10. Other, please specify |

throughout the area. Working status refers to working as a paid employee and working as self-employed; categories of non-working status includes temporary layoff, looking for work, retired, disabled, and other. Region refers to Midwest, Northeast, South, and West regions of the US. Internet access refers to access to the internet and is not specific to access at work or home nor is based on the usage.

**Table 2  Demographic characteristics of survey respondents.** Respondent characteristics by gender, age, education, ethnicity, income, marital status, metro status, region, work status, and internet access among adults 18 years and older in the United States.

| Respondent characteristics | | Do you get an influenza vaccine? ($n = 2150$) | | | | | | p-value |
|---|---|---|---|---|---|---|---|---|
| | | Yes, every year ($n = 908$) | | Yes, some years ($n = 423$) | | No, never ($n = 819$) | | |
| | | n | % | n | % | n | % | |
| **Gender** | | | | | | | | |
| | Female | 460 | 39.4% | 227 | 21.8% | 408 | 38.8% | 0.4 |
| | Male | 448 | 38.6% | 196 | 19.8% | 411 | 41.6% | |
| **Age** (years) | | | | | | | | |
| | 18–24 | 45 | 25.5% | 49 | 29.2% | 78 | 45.3% | <0.001 |
| | 25–34 | 73 | 24.0% | 82 | 28.4% | 134 | 47.6% | |
| | 35–44 | 102 | 30.9% | 75 | 23.2% | 147 | 45.9% | |
| | 45–54 | 138 | 35.5% | 86 | 22.8% | 160 | 41.7% | |
| | 55–64 | 238 | 47.8% | 78 | 14.6% | 181 | 37.6% | |
| | 65–74 | 201 | 58.3% | 45 | 13.6% | 93 | 28.1% | |
| | 75 + | 111 | 73.3% | 8 | 6.2% | 26 | 20.5% | |
| **Education level** | | | | | | | | |
| | Less than high school | 60 | 34.7% | 27 | 17.6% | 73 | 47.7% | <0.001 |
| | High school | 271 | 40.9% | 79 | 13.6% | 279 | 45.5% | |
| | Some college | 218 | 32.3% | 147 | 26.6% | 239 | 41.1% | |
| | Bachelor's degree or higher | 359 | 45.1% | 170 | 24.0% | 228 | 30.9% | |
| **Ethnicity** | | | | | | | | |
| | White, Non-Hispanic | 696 | 41.8% | 293 | 19.8% | 567 | 38.5% | 0.01 |
| | Black, Non-Hispanic | 69 | 33.2% | 37 | 18.6% | 88 | 48.2% | |
| | Hispanic | 80 | 32.8% | 54 | 24.5% | 94 | 42.8% | |
| | Other, Non-Hispanic | 36 | 37.9% | 28 | 28.3% | 29 | 33.8% | |
| | 2+ Races, Non-Hispanic | 27 | 31.5% | 11 | 14.8% | 41 | 53.7% | |
| **Income** | | | | | | | | |
| | under $10k | 24 | 27.2% | 15 | 17.0% | 50 | 55.8% | <0.001 |
| | $10k to $25k | 83 | 30.7% | 36 | 16.7% | 118 | 52.6% | |
| | $25k to $50k | 178 | 37.6% | 81 | 19.2% | 183 | 43.2% | |
| | $50k to $75k | 168 | 37.9% | 78 | 21.8% | 158 | 40.3% | |
| | $75k to $100k | 131 | 37.9% | 74 | 24.3% | 106 | 37.9% | |
| | $100k to $150k | 213 | 44.3% | 99 | 23.3% | 146 | 32.4% | |
| | over $150k | 111 | 52.8% | 40 | 19.8% | 58 | 27.3% | |
| **Marital status** | | | | | | | | |
| | Single | 302 | 33.7% | 160 | 19.9% | 374 | 46.4% | <0.001 |
| | Partnered | 606 | 43.2% | 263 | 21.6% | 445 | 35.3% | |
| **Metro status** | | | | | | | | |
| | Metro | 772 | 38.9% | 376 | 21.9% | 682 | 39.2% | 0.02 |
| | Non-metro | 136 | 39.9% | 47 | 14.9% | 137 | 45.2% | |

**Table 2** (*continued*)

| Respondent characteristics | | Do you get an influenza vaccine? (*n* = 2150) | | | | | | *p*-value |
|---|---|---|---|---|---|---|---|---|
| | | Yes, every year (*n* = 908) | | Yes, some years (*n* = 423) | | No, never (*n* = 819) | | |
| | | *n* | % | *n* | % | *n* | % | |
| **Region** | | | | | | | | |
| | Midwest | 203 | 39.0% | 91 | 19.9% | 182 | 41.1% | 0.002 |
| | Northeast | 174 | 40.3% | 81 | 19.8% | 164 | 39.9% | |
| | South | 330 | 39.5% | 127 | 17.4% | 303 | 43.1% | |
| | West | 201 | 37.2% | 124 | 28.1% | 170 | 34.6% | |
| **Work status** | | | | | | | | |
| | Not working | 454 | 46.2% | 142 | 17.5% | 296 | 36.2% | <0.001 |
| | Working | 454 | 34.1% | 281 | 23.1% | 523 | 42.8% | |
| **Internet status** | | | | | | | | |
| | No | 130 | 35.9% | 49 | 15.1% | 172 | 49.0% | <0.001 |
| | Yes | 778 | 39.9% | 374 | 22.4% | 647 | 37.7% | |

## Multivariate analysis

Statistically significant variables (*p*-value ≤ 0.05) from the bivariate analysis were included in the forward selection procedure for multivariate analysis using weighted multinomial logistic regression. We analyzed respondent demographics (age, education, income, region, work status), perceived efficacy, social influence, cost and health insurance associated with *getting the influenza vaccine* using weighted multinomial logistic regression (Table 4). We compared the responses of adults who get the influenza vaccine every year to those who get the influenza vaccine some years. We analyzed demographics (age, education, income, region, work status), perceived efficacy, health insurance, and barriers associated with *not getting the influenza vaccine* using weighted multinomial logistic regression (Table 5). We compared the responses of adults who never get the influenza vaccine to those who get the influenza vaccine some years. All statistical analysis was performed using the R software for statistical computing (*R Core Team, 2018*).

## RESULTS

### Influenza vaccination by demographics

We analyzed influenza vaccination status of adults by gender, age, education, ethnicity, income, marital status, metro status, region, work status, and internet access using Rao-Scott's chi-square test, as illustrated in Table 2. Vaccination status refers to adults being vaccinated every year, some years, or never with the influenza vaccine. We observed an association between *vaccination status* and the demographic variables of *age, education, ethnicity, income, marital status, work status, and internet access.*

A total of 73.3% of adults over 75 get vaccinated every year compared to only 25.5% of 18–24 year old adults. A total of 45.1% of adults with a bachelor's degree or higher get vaccinated every year compared with 34.7% of adults with less than a high school education. A total of 41.8% of White, non-Hispanic adults get vaccinated every year, while 53.7% of non-Hispanic 2+ races' adults and 48.2% of non-Hispanic Black adults never get

**Table 3  Perceived efficacy, social influence, herd immunity, vaccine cost, health insurance, and barriers by vaccination status among adults 18 years and older in the United States.**

| | | Do you get an influenza vaccine? | | | | | | | |
| | | Yes, every year (n = 908) | | Yes, some years (n = 423) | | No, never (n = 819) | | | |
| Variable | Response | n | % | n | % | n | % | p-value | Total (n = 2,150) |
|---|---|---|---|---|---|---|---|---|---|
| **Perceived efficacy** | | | | | | | | | |
| How effective do you think the influenza vaccine is in protecting people from becoming sick with influenza? | Very effective | 304 | 76.0% | 45 | 14.1% | 34 | 9.9% | <0.001 | 383 |
| | Somewhat effective | 435 | 41.6% | 240 | 26.9% | 285 | 31.5% | | 960 |
| | Not effective | 141 | 31.2% | 94 | 21.8% | 198 | 47.0% | | 433 |
| | It varies from season to season | 6 | 4.4% | 16 | 14.2% | 120 | 81.4% | | 142 |
| | Don't know | 19 | 7.8% | 27 | 12.5% | 180 | 79.7% | | 226 |
| **Social influence** | | | | | | | | | |
| Are you more likely to get a vaccine if others around you get a vaccine? | Yes, more likely | 254 | 64.2% | 127 | 35.8% | – | – | <0.001 | 381 |
| | No, no effect | 620 | 68.1% | 258 | 31.9% | – | – | | 878 |
| | No, less likely | 33 | 43.1% | 37 | 57.0% | – | – | | 70 |
| Are you more likely to get a vaccine if others around you do not get a vaccine? | Yes, more likely | 252 | 78.4% | 61 | 21.6% | – | – | <0.001 | 313 |
| | No, no effect | 610 | 64.6% | 294 | 35.4% | – | – | | 904 |
| | No, less likely | 37 | 33.6% | 64 | 66.4% | – | – | | 101 |
| **Herd immunity** | | | | | | | | | |
| Do you get a vaccine to protect yourself, protect others, or protect yourself and others? | Protect myself | 243 | 60.2% | 138 | 39.8% | – | – | <0.001 | 381 |
| | Protect myself and others | 653 | 68.3% | 268 | 31.8% | – | – | | 921 |
| | Protect others | 6 | 26.2% | 16 | 73.8% | – | – | | 22 |
| **Vaccine cost** | | | | | | | | | |
| How much do you pay to get an influenza vaccine? | $0 | 723 | 71.9% | 247 | 28.1% | – | – | <0.001 | 970 |
| | Less than $20 | 118 | 51.7% | 104 | 48.3% | – | – | | 222 |
| | $30 to $60 | 28 | 46.5% | 26 | 53.5% | – | – | | 54 |
| | More than $60 | – | – | – | – | – | – | | – |
| | Don't know | 34 | 35.8% | 46 | 64.2% | – | – | | 80 |
| **Health insurance** | | | | | | | | | |
| Do you have health insurance? | Yes | 887 | 41.8% | 396 | 21.2% | 708 | 37.0% | <0.001 | 1,991 |
| | No | 19 | 11.3% | 24 | 16.4% | 110 | 72.3% | | 153 |
| Are influenza vaccines covered by your health insurance? | Yes, the full cost is paid | 731 | 54.5% | 249 | 20.7% | 301 | 24.8% | <0.001 | 1,281 |
| | Yes, but only part | 61 | 36.3% | 47 | 34.6% | 44 | 29.2% | | 152 |
| | No | 19 | 36.3% | 13 | 23.6% | 23 | 40.1% | | 55 |
| | Don't know | 75 | 14.6% | 86 | 18.2% | 339 | 67.2% | | 500 |

**Table 3** (*continued*)

| | | Do you get an influenza vaccine? | | | | | | | | |
| | | Yes, every year (n = 908) | | Yes, some years (n = 423) | | No, never (n = 819) | | | | |
| Variable | Response | n | % | n | % | n | % | p-value | Total (n = 2,150) | |
|---|---|---|---|---|---|---|---|---|---|---|
| **Barriers** | | | | | | | | | | |
| | 1. The vaccine costs too much | – | – | 61 | 54.0% | 49 | 46.0% | <0.001 | 110 | |
| | 2. The vaccine is not very effective in preventing influenza | – | – | 143 | 43.4% | 196 | 56.6% | <0.001 | 339 | |
| | 3. I am not likely to get influenza | – | – | 75 | 25.2% | 203 | 74.8% | <0.001 | 278 | |
| | 4. Do not know where to get the vaccine | – | – | 14 | 35.4% | 19 | 62.6% | 0.9 | 33 | |
| What are the reasons you would not get an influenza vaccine? | 5. The side effects of the vaccine are too risky | – | – | 65 | 22.2% | 219 | 77.9% | <0.001 | 284 | |
| | 6. I am allergic to some of the ingredients in the vaccine | – | – | 23 | 40.4% | 35 | 59.6% | 0.3 | 58 | |
| | 7. I do not like shots | – | – | 70 | 26.2% | 196 | 73.8% | 0.004 | 266 | |
| | 8. I just don't get around to doing it | – | – | 183 | 48.0% | 181 | 52.0% | <0.001 | 364 | |
| | 9. I have to travel too far to get vaccine | – | – | 10 | 43.3% | 16 | 56.7% | 0.3 | 26 | |

vaccinated. A total of 52.8% of adults earning more than $150,000 get vaccinated every year compared with 27.2% of adults earning less than $10,000. A total of 43.2% of partnered adults get vaccinated every year while 46.4% of single adults never get vaccinated. A total of 46.2% of non-working adults get vaccinated every year compared with 34.1% of the working adults. A total of 39.9% of adults with internet access get vaccinated every year compared with 35.9% of adults without internet access.

Similar proportions of adults get vaccinated every year in both metro (38.9%) and non-metro (39.9%) areas. Regional vaccination status rates were also similar, with a relatively higher proportion of adults (40.3%) in the Northeast getting vaccinated every year compared with the lowest proportion of 37.2% in the American West.

### Influenza vaccination by perceived efficacy, social influence, herd immunity, vaccine cost, health insurance, and barriers

We performed bivariate analysis using Rao-Scott's chi-square test to analyze influenza *vaccination status* by *perceived efficacy, social influence, herd immunity, vaccine cost, health insurance, and barriers to vaccination*, and observed an association between them as illustrated in Table 3.

A total of 76% of adults who perceive that the vaccine is very effective get vaccinated every year compared with 31.2% who perceive that the vaccine is not effective. A total of 64.2% of adults, who are more likely to get influenza vaccination if others around them are also vaccinated, get vaccinated every year. A total of 78.4% of adults, who are more likely to get vaccinated if others around them do not get vaccinated, get vaccinated every year. A total of 68.3% of adults, who get influenza vaccination to protect themselves and others, also get vaccinated every year.

**Table 4 Demographics, perceived efficacy, social influence, herd immunity, vaccine cost and health insurance associated with getting influenza vaccine.** Demographics (age, education, income, region, work status), perceived efficacy, social influence, herd immunity, vaccine cost and health insurance associated with *getting influenza vaccine* among adults 18 years and older in the United States.

| Variable | | "Yes, every year" (versus "Yes, some years") | | |
| --- | --- | --- | --- | --- |
| | | AOR | 95% CI | *p*-value |
| **Age** (years) | (Referent: 18–24) | | | |
| | 25–34 | 1.05 | (0.54, 2.05) | 0.89 |
| | 35–44 | 1.47 | (0.78, 2.77) | 0.23 |
| | 45–54 | 1.63 | (0.89, 2.99) | 0.11 |
| | 55–64 | **3.43** | (1.9, 6.22) | <0.001 |
| | 65–74 | **4.46** | (2.3, 8.65) | <0.001 |
| | 75 + | **9.87** | (3.96, 24.6) | <0.001 |
| **Education level** | (Referent: Less than high school) | | | |
| | High school | 1.39 | (0.7, 2.76) | 0.35 |
| | Some college | 0.64 | (0.33, 1.24) | 0.18 |
| | Bachelor's degree or higher | 1.07 | (0.54, 2.09) | 0.85 |
| **Income** | (Referent: Under $10k) | | | |
| | $10k to $25k | 1.84 | (0.6, 5.66) | 0.29 |
| | $25k to $50k | 2.22 | (0.74, 6.6) | 0.15 |
| | $50k to $75k | 2.65 | (0.89, 7.9) | 0.08 |
| | $75k to $100k | 2.28 | (0.76, 6.88) | 0.14 |
| | $100k to $150k | 2.37 | (0.8, 7.04) | 0.12 |
| | Over $150k | **3.59** | (1.16, 11.04) | 0.03 |
| **Region** | (Referent: Midwest) | | | |
| | Northeast | 0.94 | (0.59, 1.49) | 0.80 |
| | South | 1.10 | (0.73, 1.64) | 0.65 |
| | West | 0.75 | (0.49, 1.15) | 0.19 |
| **Work status** | (Referent: Not working) | | | |
| | Working | 0.83 | (0.59, 1.16) | 0.27 |
| **Perceived efficacy** | (Referent: Very effective) | | | |
| | Somewhat effective | **0.28** | (0.19, 0.42) | <0.001 |
| How effective do you think the influenza vaccine is in protecting people from becoming sick with influenza? | It varies from season to season | **0.26** | (0.16, 0.42) | <0.001 |
| | Not effective | **0.16** | (0.04, 0.6) | 0.006 |
| | Don't know | **0.15** | (0.06, 0.35) | <0.001 |
| **Social influence** | (Referent: Yes, more likely) | | | |
| Are you more likely to get a vaccine if others around you do not get a vaccine? | No, no effect | **0.53** | (0.37, 0.77) | 0.001 |
| | No, less likely | **0.22** | (0.12, 0.41) | <0.001 |

**Table 4** (*continued*)

| Variable | | "Yes, every year" (versus "Yes, some years") | | |
|---|---|---|---|---|
| | | AOR | 95% CI | *p*-value |
| **Herd immunity** | (Referent: Protect myself) | | | |
| Do you get a vaccine to protect yourself, protect others, | Protect myself and others | 1.26 | (0.92, 1.72) | 0.14 |
| or protect yourself and others? | Protect others | **0.38** | (0.14, 0.99) | 0.05 |
| **Vaccine cost** | (Referent: $0) | | | |
| | Less than $30 | **0.42** | (0.29, 0.6) | <0.001 |
| How much do you pay to get an influenza vaccine? | $30 to $60 | **0.39** | (0.2, 0.77) | 0.007 |
| | Don't know | **0.25** | (0.13, 0.46) | <0.001 |
| **Health insurance** | (Referent: Yes) | | | |
| Do you have health insurance? | No | 0.71 | (0.32, 1.6) | 0.41 |

A total of 41.8% of adults with health insurance get vaccinated every year compared with 11.3% of adults without health insurance, while 72.3% of adults without health insurance never get vaccinated. 71.9% of adults who have null out-of-pocket payment for the influenza vaccine get vaccinated every year compared to 46.5% of those who pay $30 to $60. Similarly, 54.5% of adults who have the vaccine cost fully covered by their health insurance get vaccinated every year compared with 36.3% of adults who do not have the benefit.

A total of 46% of adults who are impacted by high vaccine cost never get vaccinated. A total of 56.6% of adults who perceive that the vaccine is not very effective in preventing influenza, 74.8% of adults who perceive that they are unlikely to get influenza, and 77.9% of adults who perceive that the vaccine side effects are too risky never get vaccinated. A total of 73.8% of adults who do not like shots and 52% of adults who just do not get around to getting vaccinated, never get vaccinated.

## Demographics, perceived efficacy, social influence, herd immunity, vaccine cost, and health insurance associated with getting influenza vaccination

We analyzed *demographics (age, education, income, region, work status), perceived efficacy, social influence, herd immunity, vaccine cost, and health insurance* associated with *getting influenza vaccine* using weighted multinomial logistic regression, as illustrated in Table 4. The responses by the adults for getting *influenza vaccination every year* are compared to the responses by the adults for getting *influenza vaccination some years*.

The odds of getting vaccinated every year (versus some years) were significantly higher for adults 55 years and older compared to younger adults 18–24 years, with an adjusted odds ratio (AOR) of 9.87 for adults 75 years and older, 4.46 for adults 65–74 years, and 3.43 for adults 55–64 years. Adults with annual income exceeding $150,000 had significantly higher odds (AOR = 3.59) of getting vaccinated every year (versus some years) compared to those with incomes below $10,000.

**Table 5  Demographics, perceived efficacy, health insurance, and barriers associated with not getting influenza vaccine.** Demographics (age, education, income, region, work status), perceived efficacy, health insurance, and barriers associated with *not getting influenza vaccine* among adults 18 years and older in the United States.

| Variable | | "No, never" (versus "Yes, some years") | | |
|---|---|---|---|---|
| | | AOR | 95% CI | *p*-value |
| **Age** (years) | (Referent: 18-24) | | | |
| | 25–34 | 0.92 | (0.53, 1.62) | 0.78 |
| | 35–44 | 1.20 | (0.70, 2.04) | 0.51 |
| | 45–54 | 1.09 | (0.66, 1.79) | 0.74 |
| | 55–64 | 1.40 | (0.84, 2.35) | 0.20 |
| | 65–74 | 1.04 | (0.57, 1.90) | 0.91 |
| | 75+ | 1.62 | (0.60, 4.34) | 0.34 |
| **Education level** | (Referent: Less than high school) | | | |
| | High school | 1.69 | (0.92, 3.08) | 0.09 |
| | Some college | 0.64 | (0.35, 1.17) | 0.15 |
| | Bachelor's degree or higher | 0.69 | (0.38, 1.28) | 0.24 |
| **Income** | (Referent: Under $10k) | | | |
| | $10k to $25k | 1.24 | (0.53, 2.90) | 0.61 |
| | $25k to $50k | 0.91 | (0.43, 1.91) | 0.80 |
| | $50k to $75k | 0.90 | (0.41, 1.98) | 0.80 |
| | $75k to $100k | 0.75 | (0.34, 1.65) | 0.48 |
| | $100k to $150k | 0.76 | (0.35, 1.64) | 0.48 |
| | Over $150k | 0.72 | (0.31, 1.66) | 0.45 |
| **Region** | (Referent: Midwest) | | | |
| | Northeast | 1.05 | (0.68, 1.63) | 0.83 |
| | South | 1.27 | (0.86, 1.89) | 0.23 |
| | West | **0.62** | (0.41, 0.94) | 0.02 |
| **Work status** | (Referent: Not working) | | | |
| | Working | 1.35 | (0.94, 1.93) | 0.10 |
| **Perceived efficacy** | (Referent: Very effective) | | | |
| How effective do you think the influenza vaccine is in protecting people from becoming sick with influenza? | Somewhat effective | **1.86** | (1.07, 3.24) | 0.03 |
| | It varies from season to season | **4.05** | (2.21, 7.44) | <0.001 |
| | Not effective | **10.36** | (4.58, 23.46) | <0.001 |
| | Don't know | **9.01** | (4.63, 17.54) | <0.001 |
| **Health insurance** | (Referent: Yes) | | | |
| Do you have health insurance? | No | **2.04** | (1.09, 3.83) | 0.03 |

**Table 5** (*continued*)

| Variable | | "No, never" (versus "Yes, some years") | | |
|---|---|---|---|---|
| | | AOR | 95% CI | *p*-value |
| **Barriers** | (Referent: No) | | | |
| | 1. The vaccine costs too much | **0.34** | (0.21, 0.56) | <0.001 |
| | 2. The vaccine is not very effective in preventing influenza | **0.41** | (0.29, 0.58) | <0.001 |
| What are the reasons you would not get an influenza vaccine? | 3. I am not likely to get influenza | **1.95** | (1.35, 2.81) | <0.001 |
| | 5. The side effects of the vaccine are too risky | **1.98** | (1.33, 2.94) | <0.001 |
| | 7. I do not like shots | **1.48** | (1.01, 2.16) | 0.04 |
| | 8. I just don't get around to doing it | **0.52** | (0.38, 0.70) | <0.001 |

Adults who perceive that the influenza vaccine is not effective are 84% less likely (AOR = 0.16) to get vaccinated every year (versus some years) compared to adults who perceive that it is very effective. Adults who were less likely to get vaccinated if others around them did not get vaccinated were 78% less likely (AOR = 0.22) to get vaccinated every year (versus some years), compared to the adults who were more likely to get vaccinated if others around them did not get vaccinated. Adults who get the influenza vaccine to protect others were 62% less likely (AOR = 0.38) to get vaccinated every year (versus some years), compared to the adults who get vaccinated to protect themselves. The odds of getting vaccinated every year (versus some years) were significantly lower for adults with out-of-pocket payments for the influenza vaccine compared to the adults with no out-of-pocket payments, with an adjusted odds ratio of 0.42 (58% less likely) and 0.39 (61% less likely) for adults with out-of-pocket payment of less than $30 and $30-$60 respectively.

## Demographics, perceived efficacy, health insurance, and barriers associated with not getting influenza vaccination

We analyzed *demographics (age, education, income, region, work status), perceived efficacy, health insurance, and barriers* associated with *not getting influenza vaccine* using weighted multinomial logistic regression, as illustrated in Table 5. The responses by the adults for *never getting influenza vaccination* are compared to the responses by the adults for getting *influenza vaccination some years*.

Adults who perceive that the influenza vaccine is not effective are more likely (AOR = 10.36) to never getting vaccinated (versus getting vaccinated some years) compared to adults who perceive that it is very effective. Adults without health insurance are more likely (AOR = 2.04) to never getting vaccinated (versus getting vaccinated some years) compared to adults with health insurance.

Adults who perceived low risk of influenza infection were 95% more likely (AOR = 1.95) to never get vaccinated (versus getting vaccinated some years). Adults who perceived that the vaccine side effects are too risky are 98% more likely (AOR = 1.98) to never get vaccinated (versus getting vaccinated some years). Adults who do not like shots are 48% more likely (AOR = 1.48) to never get vaccinated (versus getting vaccinated some years).

Adults who are impacted by high vaccine cost are 66% less likely (AOR = 0.34) to never get vaccinated (versus getting vaccinated some years). Adults who perceive that the vaccine

is not very effective in preventing influenza are 59% less likely (AOR = 0.41) to never get vaccinated (versus getting vaccinated some years). Adults who just do not get around to getting vaccinated are 48% less likely (AOR = 0.52) to never get vaccinated (versus getting vaccinated some years). While high cost, perception of low vaccine efficacy, and not getting around to getting vaccinated are barriers associated with never getting vaccinated, adults with these barriers are still more likely to get vaccinated some years compared to never getting vaccinated.

## DISCUSSION

### Demographics

The proportion of adults getting influenza vaccination every year increases with age, while the proportion of adults never getting influenza vaccination decreases with age. Adults with higher education levels (bachelor's degree or higher) are more likely to get vaccinated every year, while the proportion of adults who never get vaccinated increases with low education levels. While a high proportion of non-Hispanic White adults get vaccinated every year, a large proportion of non-Hispanic Black adults never get vaccinated. Adults with higher income are more likely to get vaccinated every year, while the proportion of adults who never get vaccinated increases with lower income. Partnered adults (married, living together) are more likely to get vaccinated every year, while single adults (never married, divorced, widowed) are more likely to never get vaccinated. Adults living in non-metro areas are more likely to never get vaccinated similar to adults without internet access, while adults with internet access are more likely to get vaccinated every year.

### Perceptions and socioeconomic factors

Adults who perceive that the influenza vaccine is very effective are more likely to get vaccinated every year, while adults who perceive that the influenza vaccine is not effective are more likely to never get vaccinated. A higher proportion of adults are getting vaccinated every year to protect themselves and others, and they are influenced by vaccine uptake behavior within their social networks. Adults with health insurance are more likely to get vaccinated every year, while adults without health insurance are more likely to never get vaccinated. Adults without out-of-pocket payments, as well as those for whom the cost of influenza vaccination is fully covered by health insurance, are more likely to get vaccinated every year. Adults who do not get vaccinated every year are more likely to never get vaccinated due to low perceptions of vaccine effectiveness and risk of influenza infection, high perception of risky side effects, disliking shots, and "just not getting around to do it".

### Facilitators

Facilitators for adults getting vaccinated every year in comparison to only some years include older age, perception of high vaccine effectiveness, higher income and no out-of-pocket payments. Elderly adults are likely to get the influenza vaccine every year based on their positive experience of influenza vaccination from past years, as well as act on the influenza vaccine recommendation of their physicians during their clinical visits on health
conditions that are not necessarily related to influenza (*Lu et al., 2016*). Adults with a perception of high vaccine effectiveness are likely to get the influenza vaccination based on their positive vaccine sentiment. Higher income and no out-of-pocket payments through health insurance minimize the financial burden of the influenza vaccine cost as well as the cost of taking time off from work and travel to get the vaccine (*Jerant et al., 2013*).

Adults are also influenced by vaccine uptake behavior within their social networks, including a higher likelihood of adults getting vaccinated every year if others around them are not vaccinated. This social influence can be partly explained by the Philipson model (*Geoffard & Philipson, 1997*; *Philipson, 2000*) which posits that higher disease prevalence leads to higher demand for public health interventions. An individual within a network of unvaccinated people has an enhanced risk of infection for influenza, thereby nudging the individual's choice to protect themselves by getting vaccinated.

## Barriers

Barriers for adults never getting vaccinated in comparison to only some years include lack of health insurance, disliking of shots, perception of low vaccine effectiveness, low perception of risk for influenza infection, and perception of risky side effects. Adults without health insurance are less likely to seek preventive care including influenza vaccination (*Jerant et al., 2013*). While the live attenuated influenza vaccine administered through a intranasal sprayer may appeal to adults who dislike shots in comparison to the inactivated influenza vaccine administered through an intramuscular injection, the Advisory Committee on Immunization Practices does not recommend the use of live attenuated influenza vaccine (*Grohskopf et al., 2017*).

Vaccine hesitancy and negative vaccine sentiment continues to contribute to suboptimal vaccination coverage in the United States for influenza and other vaccine-preventable diseases, posing significant risk of disease outbreaks (*Kang, Culp & Abbas, 2017*; *Kang et al., 2017*). The perception of negative sentiment in vaccine effectiveness, risk for influenza infection and risky side effects lends to vaccine hesitancy and reduces trust and confidence in influenza vaccines. Vaccine hesitancy can be addressed by leveraging the commercial and social marketing practices and principles of the four P's, namely *Product, Price, Place and Promotion* (*Nowak et al., 2015*). Specific to influenza vaccination, the *product* is the influenza vaccine, recommended annual vaccination schedule of individuals six months and older, and the act of getting vaccinated. The *price* category refers to the cost, convenience, ease of access, and perception of safety and efficacy of the influenza vaccine. The *place* category refers to the location where the influenza vaccine is administered, such as doctors' offices, clinics, and pharmacies. The *promotion* category refers to the influenza vaccine messaging and communication through posters, brochures, public service advertisements and websites, outreach through traditional news media and social media, spokespersons, and interpersonal communication including patient-provider communication. Thereby, addressing vaccine hesitancy and improving trust and confidence in vaccines by leveraging these principles and practices in commercial and social marketing has appreciable value in improving influenza vaccine acceptance (*Larson et al., 2015*; *Larson et al., 2018*; *Nowak et al., 2015*).

### Limitations

Our survey was administered to a nationally representative sample of adults 18 years and older in the United States; thereby, we are unable to extend our analysis to include children below 18 years of age.

## CONCLUSION

Health program strategies based on systems thinking focus on an ongoing, iterative learning of systems understanding, analysis, and improvement (*Swanson et al., 2012*). Influenza vaccination rates among adults in the United States are impacted by demographics, perceptions, and socioeconomic factors. Through systematic understanding, analysis, and identification of these influencing factors, this study provides evidence to improve the design and implementation of current and future influenza vaccination programs by leveraging the facilitators and addressing the barriers. Specifically, the public health implications of this study are that the influenza vaccination rates among adults in the United States can be improved towards the Healthy People 2020 target of 70% by increasing awareness of the safety, efficacy and need for influenza vaccination, leveraging the practices and principles of commercial and social marketing to improve vaccine trust, confidence and acceptance, and lowering out-of-pocket payments and covering influenza vaccination costs through health insurance.

### Funding

This study is supported by NIH/NIGMS R01GM109718 and NSF/NRT 1545362. The funders had no role in study design, data collection and analysis, decision to publish, or preparation of the manuscript.

### Grant Disclosures

The following grant information was disclosed by the authors:
NIH/NIGMS: R01GM109718.
NSF/NRT: 1545362.

### Competing Interests

The authors declare there are no competing interests.

### Author Contributions

- Kaja M. Abbas conceived and designed the experiments, analyzed the data, contributed reagents/materials/analysis tools, prepared figures and/or tables, authored or reviewed drafts of the paper, approved the final draft.
- Gloria J. Kang performed the experiments, analyzed the data, contributed reagents/materials/analysis tools, prepared figures and/or tables, authored or reviewed drafts of the paper, approved the final draft.

- Daniel Chen performed the experiments, analyzed the data, contributed reagents/-materials/analysis tools, authored or reviewed drafts of the paper, approved the final draft.
- Stephen R. Werre conceived and designed the experiments, analyzed the data, contributed reagents/materials/analysis tools, authored or reviewed drafts of the paper, approved the final draft.
- Achla Marathe contributed reagents/materials/analysis tools, authored or reviewed drafts of the paper, approved the final draft.

### Human Ethics

The following information was supplied relating to ethical approvals (i.e., approving body and any reference numbers):

The Institutional Review Board at Virginia Tech granted ethical approval for this study (IRB # 14-712).

### Data Availability

GitHub: https://github.com/gloriakang/influenza-vaccination-adults.

### Supplemental Information

Supplemental information for this article can be found online at http://dx.doi.org/10.7717/peerj.5171#supplemental-information.

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
