# Peer review of "Demographics, perceptions, and socioeconomic factors affecting influenza vaccination among adults in the United States"

_PeerJ, doi:10.7717/peerj.5171_

## Round 0.1 · original submission · Major Revisions

Overall the reviewers saw considerable merit in the article, but have concerns regarding the presentation and discussion of data. Please address their points in your revision.

·

Basic reporting

Abstract: In the conclusion, I would avoid stating “improving vaccine effectiveness”. The authors are measuring a perception and these do not necessarily meet research results on vaccine effectiveness (another research on its own). I suggest only stating the already included conclusion about the need to “increase awareness about vaccine effectiveness”.
Results: Despite the clear, unambiguous, professional language used throughout, the paper, for some variables (e.g., social influence and herd immunity items), when the outcome used is “not getting the influenza vaccine” which makes sense in terms of the study question, it has led results to some tricky double-negatives in the reporting which impair the clarity of the argument.
I am not a statistician, but it strikes me a little bit odd the presentation of results of the Rao Scott results. I understand that when the authors refer to “annual vaccination being positively correlated with age”, they are referring to the fact that the proportion of individuals reporting to have an influenza vaccine shot every year is higher in the older age groups. However, I seems to me that the term could be more appropriated employed when testing the significance of bivariate correlation and not chi square test, which tests for contingency tables.
Consider rephrasing from highest value to the lowest in the following phrase: “Differences in ethnicity were also statistically significant; a higher proportion of White, non- 128 Hispanic adults (41.8%) get vaccinated every year, whereas relatively higher proportions of non- 129 Hispanic 2+ races’ adults (53.7%) and non-Hispanic Black adults (48.2%) never get vaccinated” (Lines 127-129)

Experimental design

Study objective: I suggest making the concepts clearer about the difference between barriers to vaccination, such as financial resources and perceived barriers, which correspond to a belief or attitude (e.g., fear of side effects).

Validity of the findings

No coment

Additional comments

The paper under review tried to clarify the impact of demographics, perceptions of vaccine effectiveness and socioeconomic factors on influenza vaccination status.
The paper deals with a worthwhile topic. The manuscript is well structured, has a good rationale and broadly appropriate methods. The results are important and in the line has to what has been found in other research on the topic. However, in few minor cases, the way in which the results are presented make the paper a little bit confusing to read.

Reviewer 2 ·

Basic reporting

This is an interesting paper reporting the results of a national survey on factors affecting influenza vaccination. The survey addresses many factors previously found to impact individuals’ decision to vaccinate or not, and includes some additional factors. The figures/tables are relevant and present the results in a straightforward manner. Relevant literature is cited. There are multiple grammatical issues with the paper and in some cases the wording is confusing or misleading. The text is essentially redundant with the tables. The results are nothing more than a lengthy recitation of the data in the table. Unfortunately this section is also written such that it presents the data in a confusion manner.

Overall, the manuscript is a good start but fails to distinguish itself from prior work, neglects to indicate if it replicates or supports prior work, does not identify which factors are likely barriers to influenza vaccination or discuss what might be done about these barriers.

Experimental design

The scope of the work is within the Aims and Scope of the journal. The research questions are well defined and would provide meaningful data that could be used to improve influenza vaccine uptake. The survey of questions asked is included in the manuscript. The statistical methodology is provided and is appropriate. In the introduction, the authors have a paragraph outlining factors that have previously been studied in regards to influenza vaccination and so it is not clear how the research fills an identified knowledge gap or how the current study expands or builds on existing knowledge. Another issue is that many of the factors in the survey are potentially linked and so the effect seen by “do you have to pay to get a vaccine” may be confounded with ‘income’, which might be confounded with ‘education level’.

Validity of the findings

The Discussion is mainly a review of the results with no real discussion about what the results mean and how the data could be used. It contains essentially no discussion or conclusions, or speculation or data interpretation. The findings are not placed in the context of existing knowledge or work. Discrepancies or similarities to other studies are not presented. Interesting findings are not followed-up on or discussed. The one exception to this is the ‘Facilitators’ paragraph on line 271 where the authors attempt to dig into what drives the social effect. Unfortunately, this is limited to a single possible explanation and the section neglects to mention other obvious drivers of the results (e.g., peer pressure)

Interesting results are not followed up on or discussed. For example, those who are vaccinated every year seem more concerned about protecting themselves while those who are vaccinated in some years seem more interested in protecting others. Why is this?

Additional comments

The authors should also address the following points to improve the manuscript.

The paragraph starting on line 60 mixes factors associated with increased and decreased vaccination. It is confusing to read and ends with a statement implying that the vaccine is detrimental. The authors should organize the paragraph grouping factors positively associated with influenza vaccination and then listing those associated with decreased vaccination.

In the results section the authors bounce between associated with getting vaccinated and associated with not getting vaccinated. A good example of this is lines 125-127. At first glance the two percentages are very similar (47.7% vs 45.1%) but these numbers actually refer to different associations and the authors are trying to draw a distinction between the groups. The continual jumping (between getting vaccinated and not getting vaccinated) makes the entire section hard to follow.

In the results section, remove the words ‘in comparison’. It looks like you are comparing and contrasting age with unrelated factors.

Line 48 – insert the word ‘between’ after “Influenza has caused”.

Line 63 – awareness of what?

Line 80 – put a comma in 2,150.

Line 119 – remove the words ‘by the’ and place ‘influenza vaccine’ in parentheses.

Please define the following: metro status, region (what states are in each region), working (full-time? Part-time? Both?), internet (access at work? At home? Is access what matters or did you ask questions about usage?)

In the survey questionnaire section, the ‘health belief model’ is described but not the ‘socioecological model’. Please explain why this second model was used.

---

## Round 0.2 · accepted · Accept

Thank you for your attention to detail and thorough response to the reviewer's concerns.

·

Basic reporting

no comment

Experimental design

no comment

Validity of the findings

no comment

Additional comments

Relevant and required changes have been made to this re-submission, which now complies with the required guidelines for the journal publication.

Reviewer 2 ·

Basic reporting

The authors have addressed each of the reviewers concerns quite well. Factors that are barriers and facilitators to vaccination have been grouped together for reporting and discussion. Double-negatives and the back and forth between factors associated with vaccination and non-vaccination have been eliminated. The revised manuscript is much easier to read.

Experimental design

As with the original manuscript, the scope of the work is within the Aims and Scope of the journal and the research questions are well defined and would provide meaningful data that could be used to improve influenza vaccine uptake. The additional information on the analysis approach is also useful in clarifying the interpretation of the results.

Validity of the findings

In the revised manuscript, the findings are now placed in the context of existing knowledge or work. Interesting findings are discussed.

Additional comments

The manuscript is much easier to follow and has grouped the findings into logical categories.